# Activity and Behavioral Recognition Using Sensing Technology in Persons with Parkinson’s Disease or Dementia: An Umbrella Review of the Literature

**DOI:** 10.3390/s25030668

**Published:** 2025-01-23

**Authors:** Lydia D. Boyle, Lionel Giriteka, Brice Marty, Lucas Sandgathe, Kristoffer Haugarvoll, Ole Martin Steihaug, Bettina S. Husebo, Monica Patrascu

**Affiliations:** 1Centre for Elderly and Nursing Home Medicine, Department of Global Public Health and Primary Care, University of Bergen, Årstadveien 17, 5009 Bergen, Norway; lionel.giriteka@bergen.kommune.no (L.G.); brice.marty@uib.no (B.M.); bettina.husebo@uib.no (B.S.H.);; 2Neuro-SysMed, Department of Clinical Medicine, University of Bergen, Jonas vei 65, 5021 Bergen, Norway; kristoffer.haugarvoll@uib.no; 3Helse Vest, Helse Bergen HF, Haukeland Universitetssjukehus, Postboks 1400, 5021 Bergen, Norway; 4Department of Orthopedic Surgery, Voss Hospital, Sjukehusvegen 16, 5704 Voss, Norway; 5Department of Neurology, Haukeland University Hospital, Haukelandsveien 22, 2009 Bergen, Norway; 6Department of Internal Medicine, Haraldsplass Deaconess Hospital, Ulriksdal 8, 5009 Bergen, Norway; ole.martin.steihaug@haraldsplass.no; 7Complex Systems Laboratory, University Politehnica of Bucharest, Splaiul Independentei 313, 060042 Bucharest, Romania

**Keywords:** dementia, Parkinson’s disease, human activity recognition, sensing technology, wearables, sensors, non-motor symptoms, behavioral and psychological symptoms, systematic review

## Abstract

Background: With a progressively aging global population, the prevalence of Parkinson’s Disease and dementia will increase, thus multiplying the healthcare burden worldwide. Sensing technology can complement the current measures used for symptom management and monitoring. The aim of this umbrella review is to provide future researchers with a synthesis of the current methodologies and metrics of sensing technologies for the management and monitoring of activities and behavioral symptoms in older adults with neurodegenerative disease. This is of key importance when considering the rapid obsolescence of and potential for future implementation of these technologies into real-world healthcare settings. Methods: Seven medical and technical databases were searched for systematic reviews (2018–2024) that met our inclusion/exclusion criteria. Articles were screened independently using Rayyan. PRISMA guidelines, the Cochrane Handbook for Systematic Reviews, and the Johanna Briggs Institute Critical Appraisal Checklist for Systematic Reviews were utilized for the assessment of bias, quality, and research synthesis. A narrative synthesis combines the study findings. Results: After screening 1458 articles, 9 systematic reviews were eligible for inclusion, synthesizing 402 primary studies. This umbrella review reveals that the use of sensing technologies for the observation and management of activities and behavioral symptoms is promising, however diversely applied, heterogenous in the methods used, and currently challenging to apply within clinical settings. Conclusions: Human activity and behavioral recognition requires true interdisciplinary collaborations between engineering, data science, and healthcare domains. The standardization of metrics, ethical AI development, and a culture of research-friendly technology and support are the next crucial developments needed for this rising field.

## 1. Introduction

Currently, there are almost one billion people over the age of 60 years worldwide [1]. Increased age is associated with an increased prevalence of neurological diseases such as Parkinson’s Disease (PD) and dementia, which affect 15% of the current population [1,2,3]. PD prevalence is increasing faster than any other neurological disorder worldwide [3], and dementia is estimated to affect 139 million people globally by 2050 [2]. In addition, these diseases cost global economies trillions of dollars each year [1,2]. Sensing technologies, such as wearables, are one available solution that can help address the limitations and challenges of aging; however, they also introduce complex topics, such as the suitability of biometric applications within real-world healthcare scenarios [4].

### 1.1. Physical Activity Metrics

Physical activity metrics such as step counts, energy expenditure, awake vs. sleep time, and the intensity of various classified activities (light, moderate, and vigorous) are the most studied digital biomarkers regarding activities [5]. The accuracy challenges of commercial wearables, such as Fitbit, in the detection of physical activity biomarkers is well documented, and previous studies conclude that beyond their use for the measurement of step counts in healthy individuals, researchers should use discretion when employing the devices for healthcare decisions [5,6]. Regardless of this, activity is a highly researched metric, and decreased overall physical activity is associated with an increased mortality risk in older adults [7]. Additionally, a reduced capacity for ADLs is associated with an increased severity of behavioral symptoms [7], making both key biomarkers for older persons with neurological diseases. This umbrella review defines activities using metrics for physical activity and functional activities of daily living (ADLs), such as transitions, sitting, lying, standing, and sedentary activities.

### 1.2. Behavioral Symptoms of PD and Dementia

Although PD has classically been categorized and diagnosed based on the presence of cardinal motor symptoms, such as slowness of movements (bradykinesia), rigidity, and resting tremors, behavioral symptoms are currently being investigated as an important tool within both the prodromal (10–20 years prior to the emergence of motor symptoms) and diagnostic stages (pre-motor and motor) [8]. These behavioral symptoms go largely undiagnosed and are under-managed during disease progression. These symptoms can include constipation (50–60%), sleep disorders (60–90%), depression (35%), anxiety (40%), apathy (25–40%), hallucinations (33–42%), fatigue, pain (85%), orthostatic hypotension, urinary incontinence, and dementia [8,9]. Likewise, behavioral and psychological symptoms of dementia, also referred to as neuropsychiatric symptoms, affect up to 90% of people with dementia over the course of the illness and include agitation, anxiety, apathy, euphoria, depression, hallucinations, and sleep disturbances [10]. This review focuses on behavioral symptoms that are most commonly measured with sensing technology and associated with a diagnosis of PD or dementia, for example, agitation, anxiety, apathy, depression, and sleep disturbances. Recent studies have shown that the use of sensing technologies, such as actigraphy and machine learning predictive models, for behavioral symptoms are feasible and show positive correlations with traditional outcome measures for symptoms such as apathy and agitation [11,12,13,14].

### 1.3. Human Activity and Behavioral Recognition in People with PD or Dementia

Human activity recognition (HAR) is a fast-growing field rooted within engineering and computer science. The field has changed rapidly over the last ten years, requiring a cutting-edge and broader approach [15]. A recent scoping review concluded that use of wearables in research has increased exponentially from 2013 to 2020 and has included almost 11 million total participants [16]. Combined with other technologies, such as the Internet of Things (IoT) and Artificial Intelligence (AI), HAR can provide high accuracy, precision, and recall of activity classification [17]. HAR is classically defined as a process that identifies and classifies human activities over time based on the measurements made by digital sensing devices, wearables, or non-wearables [18]. HAR is further defined as “the art of identifying and naming activities” and expands the traditional definition to include behavioral symptoms such as agitation, sleep disturbances, pain, and apathy [19]. Common sensors utilized for HAR include an accelerometer, gyroscope, magnetometer, radar sensors (wireless), and Global Positioning Systems (GPSs).

According to the Cochrane Handbook for Systematic Reviews [20], an umbrella review, or an overview of reviews, should “use systematic methods to identify multiple systematic reviews on a related research question with the goal to extract and analyze results across outcomes”. We chose to conduct an umbrella review as there is a need to synthesize the existing literature to inform future practice and research at this pivotal point in the field of HAR. We found that many systematic reviews and surveys have been published, especially within the last 5 years, regarding the use of sensing technologies for the management and observation of activities and behavioral symptoms [21,22,23,24,25]. We chose to narrow this umbrella review to people of advanced age (>65 years) and with neurodegenerative disease (PD or dementia), as these populations continue to grow at unprecedented rates, presenting healthcare systems with immediate challenges to which these sensing technologies and techniques could be a viable solution.

### 1.4. Research Questions

What are the current methods, sensing technologies, and AI techniques being used for activity and behavioral symptom recognition in older people with PD or dementia?Are statistical analyses and study protocols within the studies heterogeneous and/or reproducible?What gaps and possibilities exist in bringing research related to the use of sensing technologies for the management and monitoring of human activities and behavioral symptoms into real-world settings and clinical practice for older adults with PD or dementia?

## 2. Methods

This umbrella review follows the standards set by Cochrane and the Johanna Briggs Institute Critical Appraisal Checklist for Systematic Reviews (JBI) [26] for methodology and reporting guidelines. We, in addition, use the rationale and guidelines for conducting an umbrella review provided by Choi et al. (2023) [27]. To identify relevant and clinically useful studies for the purposes of classifying the non-motor symptoms (i.e., agitation, apathy, and sleep disturbances) and functional activities of daily living of persons with PD or dementia, we have excluded systematic reviews primarily focused on gait and pure motor functions. In collaboration with the University of Bergen Medical Library, we developed a working search strategy, including PICO and inclusion/exclusion criteria, which are detailed in Table 1 and Table 2.

### 2.1. Search Strategy

Initial searches were conducted between 15 September and 31 October 2023 using both MeSH terms and free-text words, such as “wearable electronic devices” OR “sensor technology” OR “fitness track*” AND “complex chronic diseas*” OR “dementia” AND “human activity recognition” OR “activity recognition” AND “aged” OR “older adult*” AND “systematic review” OR “review”. The final systematic search was conducted using the Medline Ovid, Embase Ovid, Web of Science, Cochrane Library, Epistemonikos, Institute of Electrical and Electronics Engineers (IEEE Xplore), and ACM Digital Library databases on October 31, 2023. The restrictions included a publication date from 1 January 2018 to 2024. Articles before 2018 were not included, as we considered that there has been a substantial amount of newly investigated technologies over the last 5-year period, exacerbated by the pandemic and recent growth within the wearable industry [16], including research conducted using both research-grade and commercial wearables. A complete list of the search terms for each database can be found within the Appendix A (Appendix B).

In accordance with regulations set by the International Prospective Register of Systematic Reviews (PROSPERO), protocol registration was completed prior to data extraction of the included articles and accepted in PROSPERO on 9 December 2023 (CRD42023487121). Duplicates were removed using EndNote, and Rayyan [28,29] was used as a screening tool. Four reviewers (LDB, LG, MP, and LS) conducted an independent, blind screening using Rayyan based on titles and abstracts from 27 October to 10 November 2023. An unblinded assessment of full-text articles identified as “conflicts” within Rayyan was conducted by four researchers (LDB, LG, BM, and MP) between the dates of 10 November and 10 December 2023. PRISMA guidelines [30] were used for the reporting of inclusions and study selection processes.

### 2.2. Data Extraction

The data was extracted by two researchers (LDB and LG). Disagreements were settled by discussion and resolved by a third party (BM) if an agreement was not reached. The data were recorded in MS Excel spreadsheets and included categories for the author, date of publication, journal type, methods, technologies, primary articles within each systematic review (behavioral symptoms), AI, algorithms, statistical methods, thresholds, traditional measures as validators to digital biomarkers, aims, results, outcomes, datasets, and gaps in the research or future directions.

### 2.3. Risk of Bias (Quality) Assessment

An assessment of risks and bias was conducted using the JBI Critical Appraisal Checklist for Systematic Reviews and Research Syntheses [26]. The JBI consists of eleven total questions, resulting in an overall appraisal decision for (1) inclusion, (2) exclusion, and (3) seek further info (Appendix C). The results of the assessment were used to inform data extraction and synthesis and as an assessment for the overall decision and quality for inclusions (Appendix D). Three reviewers (LG, BM, and MP) were involved in the quality assessments. Disagreements between reviewers were resolved by discussion, and if an agreement was not reached, a fourth researcher (LDB) settled all disputes.

### 2.4. Data Synthesis

A total of 50% of all systematic reviews do not incorporate meta-analysis for various reasons [31,32] and none of the included systematic reviews offer a meta-analysis. Meta-analysis was, therefore, not possible in this umbrella review due to heterogeneity of the metrics and analyses used within the included systematic reviews, and a narrative synthesis of the included articles was subsequently conducted. The included content from the articles was grouped according to the following:Type of journal (technical vs. medical), authors, country, year, number and demographics of participants, and study setting (real world vs. laboratory);Relevancy to HAR used for digital phenotyping and/or classification of behavioral symptoms;Sensors, devices, AI, datasets, gold standard outcome measures, biomarkers, and validation;Reproducibility (i.e., transparency and clarity of algorithms and technical details and the inclusion of important demographic details, such as age and diagnosis);Inclusion of ethical considerations and data protection;Future recommendations from studies;Studies conducted for an advanced stage of disease or end of life;Consent procedure (informed vs. presumed).

## 3. Results

A total of 1458 articles were identified, and after the removal of duplicates (n = 392), 1066 articles were independently reviewed based on their titles and abstracts. Two additional systematic reviews were identified via snowballing, and full-text review was conducted on a total of 20 articles. Articles were excluded based on the following reasons: not a true systematic review, focus on diagnosis of disease, HAR used for only motor symptoms of PD, fall prevention or prediction specific, did not include persons with PD or dementia, focused on commercial app development, or study featured sensing technology for telehealth purposes; one article did not meet the critical appraisal tool standards and was, therefore, excluded. A total of 1058 articles were excluded, resulting in 9 included high-quality systematic reviews. Figure 1 presents the PRISMA flow diagram, detailing the process, including reasons for exclusion.

### 3.1. Included Systematic Review Characteristics

In total, within the 9 included systematic reviews (Table 3), 402 individual primary studies were assessed. The included systematic reviews represent research from eleven countries. Most of the nine included systematic reviews offered evidence for a combination of the management of activities and behavioral symptoms. We provide a summary of the nine included systematic reviews in Table 3. As recommended in a previous study [33], this umbrella review sought to describe the primary studies in a more meaningful way. We, therefore, provide the reader a detailed summary (Appendix A) of 27 primary studies [34,35,36,37,38,39,40,41,42,43,44,45,46,47,48,49,50,51,52,53,54,55,56,57,58,59,60,61] within the total 402 individual primary studies assessed, which were focused on the growing research topic of behavioral symptoms, such as agitation, apathy, and sleep disturbances, to enhance our knowledge of the metrics and methods used for the measurement of these less researched digital biomarkers for future studies. Within the 9 included systematic reviews, we identified 10 overlapping primary studies.

The sample sizes for the primary studies ranged between 1 and 2063 participants. One included systematic review [62] found that approximately 68% of publications had less than 50 participants, and, similarly, another [63] stated that studies using existing databases included an average of 10 or less participants and an average of 40 participants where ad hoc datasets were utilized. The mean ages of the included participants ranged from 22 to 95 years; five of the nine studies did not include a summary of the mean age. All included systematic reviews included people with dementia, Alzheimer’s disease, or PD as a primary or secondary diagnosis.

### 3.2. Sensing Technology, Devices, and Current Use in Research

The nine included systematic reviews combined the findings for accelerometry for the detection of both activities and behavioral symptoms [62,63,64,65,66,67,68,69,70]. In addition, non-wearables [62,63], pressure sensors [63,68], robots [63], smart devices [62], GPSs [63,65], triaxial and inertial sensors (including accelerometers, gyroscopes, and magnetometers) [62,63,68,69], and ambient home sensors [62,64,67] also made the stage; combining multiple wearable sensors (wrist and low back placement being the most common) and video or diary input for the analysis and confirmation of data (i.e., agitation events) was noted regarding efforts for the classification of activities and behavioral symptoms [62,67,68] (Table 3).

The included systematic reviews [65,66] describe the use of sensing technologies to measure the volume, intensity, pattern, and variability of physical activities of persons with dementia, stating that this is currently the most common use of accelerometry (actigraphy) within this population of patients. In agreement with one included systematic review [69], we found that the motor symptoms of PD are well studied and that research on sensing technologies for the detection of behavioral symptoms is currently growing. An example of this primary research focus on motor symptoms of PD within our findings is the included review [70], which investigated home monitoring possibilities for persons with PD using sensing technologies but included only one primary study dedicated to behavioral symptoms within their review, emphasizing the need for more synthesized literature on this topic (Appendix A).

### 3.3. Traditional Outcome Measures

The traditional “gold standard” assessment tools most frequently used within the included systematic reviews were Cohen-Mansfield Agitation Inventory (agitation) [71], Mini-Mental Status Exam (cognition) [72], Unified PD rating scale (staging of disease and function) [73], Montreal Cognitive Assessment (cognition) [74], and the Neuropsychiatric Inventory (behavioral symptoms) [75]. This umbrella review had similar findings to one of the included articles [68] and found that 22% of the included systematic reviews did not include information regarding gold standard assessment comparisons and that 67% included some information but were incomplete. Digital biomarkers within the studies were also vast, with the most utilized being physical activity parameters, such as the sleep vs. wake time, step counts, activity counts, intensity of activities (low–high), activity amplitude, and sedentary time, and the classification of activities of daily living, such as upright posture, sitting, standing, and walking (Table 3). Sleep disturbances and agitation were the most investigated behavioral symptoms.

**Table 3 sensors-25-00668-t003:** Included systematic reviews, properties, and characteristics.

Author and Country	Ardelean and R. Redolat (2023) [64], Spain	Breasail et al. (2021) * [66], United Kingdom	Esquer-Rochin (2023) * [63], Mexico
**Aim and Demographics**	To determine how technology can help to improve the support for behavioral and psychological challenges of dementia.	Description of outcome measures and the identification of studies that show a relationship between neurodegenerative disease and digital biomarkers.	To investigate the state of the art of the IoT in dementia.
** *Mean age (years)* **	60–95	28.3–85.5 Participants < 63 years were included as healthy controls	Not included
** *Number of participants* **	9–455	5–455	10–42
** *Number of included studies* **	18	28	104
**Included activities/behaviors**	Behavioral symptoms: Behavioral and psychological symptoms of dementia (BPSD) in persons with Alzheimer’s disease	Activity and behavioral symptoms: Physical activity, sedentary behavior, sleep disturbance, rest-activity patterns, and motor symptoms of PD	Activity and behavioral symptoms: ADLs, agitation, and wandering
**Sensing Technologies**	wearable triaxial accelerometer, daysimeter (rest–activity, sleep), GPS, non-wearable actigraphy device (under mattress, sleep), wrist actimetry, mobile phone, robots	GPS, sensors, and accelerometers	RF devices, Beacon GPS, Inertial devices, smartphones, glasses and watches, binary proximity sensors, ambient temperature, smart meter, video, and neuroimaging devices.
**Observational period**	3 days, 3–4 weeks, 3 months, 1–5 years (most common being 3 months)	24 h-3 months; most common 7 days	Not included
**Algorithms and Artificial Intelligence**	Not included	Not included	Random forest, decision trees, support vector machines, k-nearest neighbors, and (deep) neural networks.
**Digital Biomarkers**	Psychological symptoms: depression, anxiety, and apathy Behavioral symptoms: sleep disturbances, agitation, and wandering	Step count, time spent in physical activity, number of bouts, MET, awake/sleep time, time spent sedentary, trip frequency, duration outside home, walking duration, aggregation of velocity data into 60 s epochs, activity levels, algorithm classification of upright posture, sitting, standing, walking, walking speed cut-offs for PD, gait/motor PD, activity intensity and levels, and sleep activity	Activities of daily living, speech/voice, location/GPS, vital signs, brain/neurological- related variables, position within a room, wandering, or agitation-related activities.
**Included Comparative Measures**	MMSE, NPI, NPI-NH, CDR, IQCODE, VAS, ADL, CADS, CMAI, QUALID, FAB, DEMQOL, EQ-5D-5L, QUIS, S-MMSE, MSPSS, HDRS, FCSRT, FAST, TMT-A/B, STROOP Test, DSST, AI, NOSGER, QOL-AD, TBA, CGA, CDT, ET, STAI, HADS, NQOL, MDS, RUDAS, AS, CAM, GDS, RAID, CSDD, ACE-R, TELPI, AIFAI, WHOQOL-OLD, BARS, APADEM-NH, PSQI, AES,MDS-ADL, DAD, CERAD-NB, WMS-III, CFT, DSMT, DSST, video	ALSFRS-R, MoCA, PDQ, PASE, LSA	Not included
**Results and Conclusions**	Technologies can help people living with AD and dementia.	Accelerometers utilized more than GPS in the literature (27/28 included primary studies)	IoT in targeted dementia studies looked at biomarkers for ADLs, location, presence, vital signs, brain related variables, position within a room, and wandering.
	Technologies are useful in the management and control of BPSD.	Seven days was most common measurement time	IoT was used for caregivers of people with dementia, people with dementia, healthy older adults, medical experts, and IoT experts.
	Symptoms best managed with the help of technologies are depression, sleep disorders, anxiety, apathy, motor activity, and agitation.	Quantification of physical activity in persons with neurodegenerative disease using accelerometers can potentially provide continuous monitoring of behavioral patterns and sleep activity.	IoT was used for the detection of disease, monitoring of patients, localization of patients, assistance to patients, and cognitive training.
	Benefits of technology use for people with AD and dementia: higher quality of life, decreased expenses, better care by health professionals, and better communication and connection between professionals, patients, and families.	Accelerometry may be an objective method to establish disease progression/staging.	Qualitative and quantitative methods were used.
	Technology can revolutionize the management of BPSD.	Remote assessments using sensors for Timed Up and Go (TUG) are possible.	Data used were ad hoc and existing datasets.
	More studies of improved quality are needed to generalize optimal use and application of these technologies.	Placement of sensors, especially with persons with PD, is important.	IoT devices included wearables and environmental sensors (inside/outside).
		Major limitations include battery life, practicality of daily use for persons with dementia, acceptability, size, shape, materials used, and placement of sensors.	Both supervised and unsupervised machine learning approaches were used, with 73% being supervised.
		Need for standardization of data processing methods and algorithm transparency.	Mild cognitive impairment (MCI) was the most studied stage of dementia, followed by Alzheimer’s disease; PD was the least studied disease.
			Detection of disease was the most studied objective, followed by monitoring of patients.
			15 identified data sets within the included papers; only 5 related to people with neurodegenerative disease.
			Top 5 future suggestions: collect more data, real-world settings, validation, machine learning algorithms, and creating functionality
**Author and Country**	**Johannson et al. (2018) [67],** **Sweden**	**Khan et al. (2018) [68],** **Canada**	**McArdle et al. (2023) [65],** **United Kingdom, New Zealand, and Australia**
**Aim and Demographics**	Synthesis of knowledge from quantitative and qualitative clinical research using wearable sensors in epilepsy, PD, and stroke.	Identification of studies that use different types of sensors to detect agitation and aggression in persons with dementia.	To understand habitual physical activity participation in people with cognitive impairment, identify metrics used to assess activity, describe differences between people with dementia and healthy controls, and make future recommendations for measuring and reporting activity impairments
**Mean age (years)**	34–71	74.3–85.5 * 7/13 studies included no age information	22–84; majority 63–84
**Number of participants**	5–527	6–110	7–323
**Number of included studies**	56	14	33
**Included activities/behaviors**	Activities and behavioral symptoms: Physical activity metrics, walking, sleep disturbances, and seizures	Behavioral symptoms: Agitation and aggression	Activity: Physical activity metrics
**Sensing Technologies**	accelerometry, gyroscope, wearables	accelerometry, gyroscope, wearables, camera, and ambient sensing modalities	wearables, ambient home-based sensors, and accelerometer (most commonly wrist worn or low back)
**Observational period**	1–9 days lab setting 8 h–7 days free living	Timeframe not detailed for all studies; 3 h, 48 h, 5–7 days	Most common was a 7-day protocol varying from 2 days to 3 months (capturing weekdays and weekends)
**Algorithms and Artificial Intelligence**	Commercial algorithm (Parkinson’s KinetiGraph), time–frequency mapping, Fast Fourier transformations, support vector machines, iterative forward selection algorithm, linear discriminant analysis, discriminant analysis to determine the threshold of mean duration of immobility, combined axis rotations, power spectrum area and peak power, root mean square, mean velocity, frequency, and jerk.	Rotation forest, Hidden Markov Models, Support vector machines, Bayesian Network, and Time–frequency analysis	Not included
**Digital Biomarkers**	Step counts, energy expenditure during walking, tremor, dyskinesia, postural sway, spatiotemporal gait, medication evoked adverse symptoms, tonic–clonic seizures, non-epileptic seizures, motor seizures, sleep disturbance, upper extremity activity, and walking.	agitation and aggression	Steps per day, outdoor time, activity counts, low-vigorous activity (METS), total movement intensity, mean vector magnitude of dynamic acceleration per day for total behavior, expressed relative to gravitational acceleration, time spent walking, time of day activity (day, night, etc.), relative amplitude (higher amplitude indicates stronger rhythm; rest–activity), hour to hour and day to day variability, root mean square difference, interindividual variability, intra-daily stability and variability, and COV of daily activity.
**Included Comparative Measures**	Video, gait analysis, functional activities analysis, UPDRS III, CDRS, mAIMS, MBRS, GAITRite, PIGD, PDQ-39, MiniBEST, SF-36, commercial system (SAM, PAL, and TriTrac RT3), commercial system (sensing stylus, Actical, ActivPAL, Vitaport and Kinesia), NIHSS, NEADL, FMA, ARAT, WMFT, stroke ULAM, MAL, MAL-26, AAUT, BBS, FIM, mRS, and 6 MWT	CMAI, MMSE, DSM-III-R, ABS, NPI, SOAPD	MoCA
**Results and Conclusions**	Wearables were used in a laboratory, hospital, and free living	The most prevalent behavioral and psychological symptoms are apathy, depression, irritability, agitation, and anxiety.	Represents the literature from 16 countries.
	Good agreement with step count for patients with stroke	Stage of disease (Alzheimer’s) affected activity levels: early had increased activity before sunset, middle stage had increases at sunset, and advanced stage had more activity after sunset.	>50% of participants were female.
	Moderate to strong agreements between dyskinesia and clinical rating for persons with PD.	Moderate but highly significant correlation between CMAI scores and actigraphy data.	61% of studies were cross-sectional; 33% used data from RCTs.
	Good agreement between sway and spatiotemporal gait measures for persons with PD.	Significantly lower approximate entropy (fractal dimension ratio) during a 24-h period and at night for people with aggression.	Most studied stage of disease with MCI.
	Video assessment was used to confirm accuracy of device.	No significant correlation between agitation and motor activity (wrist actigraphy).	94% of studies included wearables.
	Accelerometry measures from 1 to 9 days	High levels of activity during the day for patients with high CMAI and low MMSE scores.	Most used device was an accelerometer (wrist).
	Wearing time in free living studies was 8 h to 7 days.	Strong correlation between mean motor activity of persons with dementia and the CMAI scores.	Most common length of observation was 7 days.
	Video electroencephalography, clinical scales, and polysomnography were used as “gold standard” references to validate biomarkers from the wearables.	Significant correlations between sensor variables and CMAI (morning and afternoon), and Aggressive Behavior scale (ABS) (morning, afternoon, evening)	Very light physical activity; 145 to 274 counts per minute
	Movement patterns for seizures (epilepsy) were detected via accelerometry 95% of the time (verified with video).	Computer vision, multimodal sensing (fusion architecture), and machine learning techniques used.	Light to moderate physical activity; 274 to 597 counts per minute
	Detection sensitivity for convulsive seizures was 90–92%.	8 studies show correlation between actigraphy and agitation in persons with dementia.	Moderate-vigorous physical activity; >3 METS and >587 to 6367 counts per minute.
	Differentiation of psychogenic non-epileptic seizures from epileptic seizures was 93–100% sensitivity.	1 study used a video camera to identify agitation.	3 studies classify vigorous activity as >6 METS and counts per minute between 5743 to 9498.
	Upper extremity measures discriminated well between those with stroke, healthy individuals, and between impairment levels.	6 studies used multimodal sensors.	Counts per minute for persons with dementia were in the very light or light to moderate ranges.
	Poor to moderate correlation in free living environment for step and activity metrics between accelerometry and the unified Parkinson’s disease rating scale.	8 studies used various statistical and machine learning methods; the other 6 did not use this.	Interpretation of findings is limited by the lack of standardization (metrics).
	Adherence to wearables was moderate (53–68%)	7 studies include demographic information (gender/age).	>50% did not report information on the validity of the devices.
	Challenges of using wearables included acceptability and integration into daily life, lack of confidence in technology, and the need for tailoring to improve use friendliness.	10 studies used various clinical assessments to verify the results from actigraphy parameters.	44 total metrics were captured across the studies.
		7 studies performed in a naturalistic setting.	Metrics related to volume and intensity were most used.
		Only 4 of the studies discuss ethics.	Lack of information regarding demographics.
		Validation of technologies is critical.	
**Author and Country**	**Morgan et al. (2020) [62],** **United Kingdom**	**Mughal et al. (2022) * [69],** **Pakistan, United Kingdom, Saudi Arabia, and Slovakia**	**Sica et al. (2021) [70],** **Ireland**
**Aim and Demographics**	Provide an overview of what technology is being used to test outcomes in PD in free living participants’ activities in a home environment.	To present different techniques and for early detection and management of PD motor and behavioral symptoms using wearable sensors.	To investigate continuous PD monitoring using inertial sensors, where the focus is papers with at least one free living data capture unsupervised (either directly or via videotapes).
** *Mean age (years)* **	Not included	Not included	Not included
** *Number of participants* **	Not included	4–2063	1–172
** *Number of included studies* **	65	60	24
**Included activities/behaviors**	Activity and behavioral symptoms: Physical activity and sleep disturbances	Behavioral symptoms: Sleep dysfunction, depression, impulse control, and motor symptoms of PD	Activity: ADL (transitional), physical activity, and motor symptoms of PD
**Sensing Technologies**	Various sensing technologies	Inertial sensors (IMU), triaxial accelerometers, gyroscopes, and magnetometers. Micro-electro-mechanical system (MEMS), necklace, and barometer. Cameras: Zenith and Kinect. Capacitive pressure sensor. Surface EMG. IMUs; Mechanomyography, flex and light sensors, Ambulatory Circadian Monitoring (ACM), polysomnography, smart toilet, and EEG sensors	accelerometer, gyroscope, and magnetometers
**Observational period**	2 weeks or less (majority) 10 studies; up to 1 yr 3 studies; multiple measurements over time	Not included	Hours-14 days
**Algorithms and Artificial Intelligence**	Not included	Not included	Artificial Neural Networks, Fuzzy logic algorithms, linear regression, and Support Vector Machine, Diverse Density, Expectation Maximization version of Diverse Density, Discriminative variant of the axis-parallel hyper–rectangle, Multiple–Instance learning, and k–Nearest Neighbor
**Digital Biomarkers**	Tremor, gait, typing, medication on/off, sleep, physical activity, bradykinesia, dyskinesia, skin temperature, light exposure, posture, falls, and activities of daily living (majority gait and motor-related symptoms)	Motor symptoms: tremor, bradykinesia, rigidity, and freezing of gait.Behavioral and other symptoms: gastrointestinal problems, sleep disfunction, impulse control disorder, depression, and physical activity metrics.	Gait impairments, step counts, intensity and volume of activities, kinematics, bradykinesia, tremor, dyskinesia, and on/off state episodes
**Included Comparative Measures**	UPDRS, PDQ-8, PDSS, FIM, PSQI, NADCS	UPDRS, HY, TUG, polysomnography, EEG	UPDRS, PASE, and symptom diaries
**Results and Conclusions**	Clinical rating scales such as the MDS-UPDRS, are currently the gold standard to measure disease severity in PD; however, they are highly subjective, non-linear, and display a “floor effect” during early-stage disease.	Behavioral symptoms of PD are often ignored.	All studies included use of accelerometry.
	68% of the included studies had sample sizes less than 50.	Behavioral symptoms are correlated with motor symptoms.	Most studies used commercial sensors/wearables.
	Almost 20% had fewer than 10 participants.	The most common current techniques used to assess PD are the UPDRS, HY, and TUG.	5 of 24 studies used prototype sensors.
	88% were observational studies	These assessments are subjective and time consuming.	Gait and motor symptoms mostly studied.
	A home-like environment was used in the majority of studies; however, most did not conduct the research in the actual home of the participants.	Use of wearables in research has grown tremendously since 2019 and is expected to continue growing.	10 of 24 studies examined symptoms and side effects of treatments.
	Duration of observation was 2 weeks or less	Many studies propose the use of sensors for both motor and behavioral symptoms.	Multiple sensors used with most common placement on lower extremities (motor).
	10 studies had an observation time of >1 year	These techniques are not yet common clinically.	Persons with PD take smaller turns when walking.
	Biomarkers included gait, tremor. Physical activity, bradykinesia, dyskinesia and motor fluctuations, falls, posture, typing, sleep, and ADLs.	Areas of application include diagnosis, tremor, body motion (motor) analysis, motot fluctuations (on–off phases), and home long-term monitoring.	No correlation between PASE and steps taken or time spent in moderate to vigorous activity.
	Most common devices were wearables and smartphones.	Other areas include fall estimation and prevention, fall risk, and freezing of gait.	Frequent sensor-derived measures were successfully able to predict future falls.
	Multimodal sensors were utilized in many of the studies.	PD motor use include symptoms of gait, tremors, bradykinesia, and dyskinesia.	Ad hoc hardware and on-board algorithms could enhance real-time feedback.
	54% of the studies mentioned validation of the technologies using video, clinical observation, participant diaries, comparison with a clinical rating scale, instrumented walkway, motion analysis, telephone calls, polysomnography, and sleep respiration monitor.	Practical issues of clinical use include cost and design.	“Black box” software and manipulation of raw data should be avoided.
	Testing the technology against itself using test–retest repeatability and responsiveness may be the best way to validate results	Motor symptoms have been a key area of interest.	Comfort of use, set-up, instructions for use, support, aesthetics, and display should always be considered.
	12 studies did not include clinometric properties, and the others used diverse design, methods, sample sizes, and statistical analyses.	Commercial wearables were included in 3 of the studies but will most likely represent greater percentages within research in the future, fueled by the pandemic.	
		Behavioral symptoms studied included depression, impulse control disorders, sleep dysfunction, and gastro-intestinal problems.	
		Most common to use 3 or greater sensors.	
		There is very little research on behavioral symptoms.	
		The future of wearables is in testing in real-world environments.	

* Published in technical journals. Acronyms: 6 MWT: 6 min walk test; AAUT: Actual Amount of Use Test; ACE-R; Addenbrooke Cognitive Examination-Revised; ADLs: Activities of Daily Living; AES: Apathy Evaluation Scale; AI: Apathy Inventory; AOAFAI: Adults and Older Adults Functional Assessment Inventory; ALSFRS-R; ALS Functional Rating Scale-Revised; APADEM-NH: Apathy Scale for Institutionalized Patients with Dementia, Nursing Home version; ARAT: the Action Research Arm Test; AS: Apathy Score; BARS: Brief Agitation Rating Scale; BBS: Berg Balance Scale; CADS: Changes in Advanced Dementia care Scale; CAM: Confusion Assessment Method; CDR: Clinical Dementia Rating; CDRS: Clinical Dementia Rating Scale; CDT: Clock Drawing Test; CERAD-NB: Consortium to Establish a Registry for Alzheimer’s Disease Neuropsychological Battery; CFT: Category Fluency Test; CGA: Comprehensive Geriatric Assessment; CMAI: Cohen-Mansfield Agitation Inventory; CSDD: Cornell Scale for Depression in Dementia; DAD: Disability Assessment for Dementia; DEMQOL: Dementia Quality of Life; DSM-III-R: the Diagnostic and Statistical Manual of Mental Disorders; DSMT: Digital Span Memory Test; EEG: electroencephalogram; EQ-5D-5L; EuroQol-5 Dimensions; ET: Emotional Thermometer; FAB: Frontal Assessment Battery; FAST: Functional Assessment Staging Test of Alzheimer’s Disease; FCSRT: Free and cued selective reminding task; FIM: Functional Independence Measure; FMA: Fugl-Meyer Assessment; FRT: Functional Reach Test; GDS: Geriatric Depression Scale; GPS: Global Positioning System; HADS: Hospital Anxiety and Depression Scale; HDRS: Hamilton Depression Rating Scale; IQCODE: Informant Questionnaire on Cognitive Decline in the Elderly; LSA: Life Space Assessment; MAL: the Motor Activity Log; MAL-26: Motor Activity Log-26; MBRS: Modified Bradykinesia Rating Scale; MDS: Modified Depression Scale; MDS-ADL: Minimum Dataset Activities of Daily Living Scale; MiniBEST: Mini-Balance Evaluation Systems Test; MMSE: Mini-Mental State Examination; mRS: Modified Rankin Scale; MSPSS: Multidimensional Scale of Perceived Social Support; NADCS: Nocturnal Akinesia Dystonia and Cramp Score; NEADL: the Nottingham Extended Activities of Daily Living Questionnaire; NIHSS: the Nation Institutes of Health Stroke Scale; NOSGER: Nurses’ Observation Scale for Geriatric Patients; NPI: Neuropsychiatric Inventory; NPI-NH: Neuropsychiatric Inventory Nursing Home; NQOL: Neuro Quality of Life in Neurological Disorder; PASE: Physical Activity Scale for the Elderly; PD: Parkinson’s Disease; PIGD: Postural Instability and Gait Disorder subscore; PSQI: Pittsburgh Sleep Quality Index; QOL-AD: Quality of Life Alzheimer’s Disease; QUALID: Quality of Life Scale for Severe Dementia; QUIS: Quality of Interactions Schedule; RAID: Rating for Anxiety in Dementia; RUDAS: The Rowland Universal Dementia Assessment Scale; SF-36: Short-Form Health Survey; S-MMSE: Severe Mini-Mental State Examination; STAI: State–Trait Anxiety Inventory; S-ULAM: Stroke Upper Limb Activity Monitor; TBA: Tinetti Balance Assessment; TELPI: Teste de Leitura de Palavras Irregulares; TMT-A/B: Trail Making Test; DSST: Digital Symbol Substitution Test; UPDRS III: Unified Parkinson’s Disease Rating Scale; VAS: Visual Analog Scale (pain); WHOQOL-OLD: World Health Association Quality of Life—Older Adults module; WMFT: the Wolf Motor Function Test; WMS-III: Wechsler Memory Scale-III.

### 3.4. Study Protocols and Methods

The protocols for data collection using sensing technologies varied greatly from several hours to as long as one year, with the most common metric being 7 days [64,65,66,67] followed by 48–72 h [64,65,67,69]. In 33% of the included systematic reviews [62,67,68], the protocol was either not mentioned or not given in a full description for each study. The statistical analyses, protocols, and methods used within the articles investigated in the systematic reviews were heterogeneous, and subsequently, meta-analysis was not possible or not performed. A summary of the statistical analyses utilized within the included primary studies, concentrated on articles covering behavioral symptoms, is provided in the Appendix A (Appendix A).

### 3.5. AI and Algorithms

A wide range of algorithms and AI techniques are mentioned in the included studies, among which are fuzzy logic, time–frequency analysis, forward feature selection, random forest, k-nearest neighbors, support vector machines, and artificial and deep neural networks [62,66,67,69]. Support vector machines were the most prevalent across all studies, while 55% of the studies [33,63,64,65,69] did not provide information on algorithms/AI for extracting activity/behavior information from sensor data. One of the included systematic reviews [70] provided a robust synthesis of AI, signal processing information, and performance validity of the included primary studies, reporting that algorithm-based classification methods using digital biomarkers for bradykinesia, transitions (turning), gait, and tremor resulted in moderate to high accuracy and validity compared to corresponding symptom diaries, and reflected upon the potential of AI and digital biomarker-based algorithms in efforts to enhance the amount of meaningful data gathered using commercial devices in a real-world setting [70].

### 3.6. Datasets

One [63] out of the nine included systematic reviews highlighted current datasets that were used within simulated, laboratory-based studies and included a total of 104 primary articles, providing a thorough description of the current datasets. Of the existing mentioned datasets, ADNI, PD Telemonitoring, AZTIAHO, Parkinson’s dataset, and Daphnet include participants that have a diagnosis of Alzheimer’s or PD [63]. Three others additionally include older adults: OASIS, CASAS, and PUCK; however, details of the demographics are not clearly stated [63]. Only 2 out of the 15 total datasets were directly applicable to use for HAR for persons with PD or dementia. A description of all included datasets can be found in Appendix E.

### 3.7. Quality and Bias Assessment

We conducted a critical appraisal of quality and potential biases using the Johanna Briggs Institute (JBI) Critical Appraisal Checklist for systematic reviews and found a score of eight out of eleven (moderate–good) for all included studies, with the exception of one umbrella review, where only 6 out 11 questions were answered with a “yes”. This umbrella review was excluded based on its JBI score. A detailed list of questions on the JBI and conducted review can be found in the (Appendix D and Appendix E). Assessment scores were most affected because the studies did not provide a clear statement as to whether a critical appraisal was performed and what tools were used.

### 3.8. Recommendations from Included Systematic Reviews

The future perspectives provided by the articles published within the technical vs. medical journals differed greatly. An example can be seen in the summary of suggestions in one of the included technical systematic reviews [63], concluding that the collection of more data is the highest priority among the 104 technical primary papers included in their systematic review. In contrast, the highest-ranking future priorities within the articles published within the medical journals were ethical concerns, the development of best practices for data management, and technology that is best suited to the participants and research needs (validation of technology) (Appendix F). The included systematic reviews [64,68] discuss ethical and safety issues, including the unforeseen consequences of technology, and conclude that these topics should be highlighted in future research. The results emphasize the need to take a “leap of faith” in the application of HAR and similar technologies for clinical trials and their implementation into real-world settings. Commercial vs. research-grade sensing technologies are further discussed [66], including the need to best discriminate between which is most appropriate for future use in studies and real-life applications, emphasizing the need to improve and validate devices and algorithms.

## 4. Discussion

This umbrella review sought to answer several research questions and found that (1) the methods, sensing technologies, algorithms and AI techniques being used in HAR for persons with PD or dementia are diverse, and the standardization of the metrics and strategies in the field, across disciplines, is a necessary step forward to allow for the potential application of HAR in clinical and real-life settings; (2) statistical analyses and study protocols are heterogeneous and introduce potential bias, decreasing the generalization and reproducibility of the study results; and (3) there are many opportunities for furthering research withing the field of HAR for persons with PD and dementia, including the topics/gaps of behavioral symptom management, the use of HAR throughout the progression of disease, including the end of life, improved data mining techniques and management of big data (minimalization), the validation of technologies, and encouraging cross-collaboration and true interdisciplinary work.

The included systematic reviews [64,68,70] found that the most common current use of HAR for behavioral symptoms is for the detection and management of depression, apathy, agitation, and sleep deprivation; however, we agree most with the systematic review [69] that stated that behavioral symptoms are often ignored and currently not well studied. We recommend that a future systematic review be performed on the latest HAR methods for behavioral symptoms as a continuation and update of this topic, especially considering the rapidly evolving nature of the field.

The sample size is a common theme throughout the included systematic reviews. We will note that when it comes to the use of sensors, dataset size assessments are fourfold: unique sources (number of participants), datapoints (number of measurements), variety (number of action types), and modality (number of sensor types). Digital sensors usually generate a significant number of datapoints, from 24 per day (e.g., actigraphy measures) to approx. 8.3 million per day (e.g., three-axis accelerometer at 32 Hz).

### 4.1. Under-Representation and Ethical Considerations

Many people with dementia and other advanced neurological conditions are excluded from research based on structural discrimination, measures designed to protect them from harm (i.e., informed consent), or inadvertently through a lack of awareness of participants’ needs and difficulties [76]. Study demographics, including the age and disease stage, can have a big impact on results [77,78]. In 2020, Alzheimer Europe published a report titled *Overcoming ethical challenges affecting the involvement of people with dementia in research: recognizing diversity and promoting inclusive research* [79]; the report states that ethics in dementia research extends from traditional standards of “do no harm” to include empowerment, rights, respect, equity, and well-being and emphasizes that appropriate adaptations must be made to ensure that people with dementia have the same opportunities to take part in research. Likewise, future studies involving these vulnerable participants should consider possible unforeseen challenges and consequences of introducing these devices at later stages of disease, and acceptance should be addressed within the studies, as emphasized within the included systematic reviews [67].

The important topic of ethics related to the use of sensing technologies with vulnerable groups of patients is brought to light and is a gap within the literature, as stated in one included systematic review [68], since only 7% of the included studies addressed ethical challenges. This finding is similar to our findings across all nine included reviews. A recent study [80] has explored the “disruptive power” of AI in elderly care and identified four main risks in the use of AI and phenotypes for detecting and managing neurodegeneration in older adults: depersonalization, discrimination (including ageism), dehumanization, and disciplining (i.e., enforcing norms) through the collection of big data. The author discusses issues such as the use of AI for clinical decision making and the reliance on algorithms for the prediction of health outcomes, assessment of risks, and choice of appropriate interventions, which have a large impact on clinical responsibility, accountability, and trust [80].

### 4.2. Future Directions

We identified several intersections between the medical and technical literature included in this umbrella review for future research directions: more research in real-world settings, the validation of technologies for the monitoring of symptoms of PD and dementia, more research conducted at late stages of neurodegenerative diseases, and adaptive technology to individual progression (precision models). The standardization of metrics, protocols, statistical methods, and use of AI and algorithms is necessary for real-life, real-time applications of sensing technologies. A novel paradigm was introduced [81] that the authors labeled as “explainable digital phenotyping” influenced by the brain, body, and social behavioral decline, being based on the validation of digital biomarkers and reliant on multidisciplinary co-creation combining AI and knowledge from healthcare professionals; the authors highlight the complexities of using digital sensing technologies as an unbiased data-driven approach for behavioral quantification, including the need for empirical consensus on standardization, reliability, and interpretation.

A recent article [82] describes dataset information related to “REal world Mobility Activities in PD” and details a total of twenty existing datasets, with only two including participants >44 years of age. The article further provides foundational research for future datasets with a detailed opened dataset named REMAP featuring transitional ADLs and participants with PD (61.25 years ±8.5) [82]. Conclusions from previous studies [82,83] support our findings [63] that current datasets are largely unrelated to persons with PD or dementia, and future work should provide more robust datasets featuring transparent representation of older people with neurodegenerative diseases.

One included systematic review [65] revealed that over 50% of the included primary studies did not include information about the validation of the technology used. A future suggestion revealed in our findings [62] was that testing the technology against itself using test–retest repeatability may be the best way to validate results in future studies. Traditional methods for validation, such as comparison to a gold-standard outcome measurement, may not correlate to the measurements taken with the device, and this may not always be a negative scenario. Measurements taken through HAR sensing methods should, in theory, be able to pull out details that subjective traditional outcome measures, such as the NPI, do not capture. This makes it difficult to conduct validation comparisons in the traditional sense. With the challenge of rapid obsolescence of new technologies, future studies may need to shift validation efforts away from specific device brands and models and towards the characteristics of the devices that are the best fit for patient care scenarios and develop new methods for validation.

## 5. Limitations and Strengths

This systematic umbrella review has several limitations. Gaps in the evidence are possible should the included systematic reviews not comprehensively cover the intended topic, which was found to be true in this case, as most of the included reviews reported results for our topic within a broader context, encompassing both motor, activity, and behavioral uses for HAR. As a result, we identified a gap in the literature surrounding the topic of HAR for the management and monitoring of behavioral symptoms. To address this gap, we included, for the reader, further analysis of the primary studies from within the included systematic reviews, which were directly related to behavioral symptom use for HAR (Appendix A) in efforts to expand the knowledge on this growing topic of interest. The overlap of the included primary studies was assessed and reported to further reduce bias. The validity of any umbrella review depends highly on the quality of the included studies and existing systematic reviews. To assess the quality and further reduce bias, three researchers performed a critical appraisal of all included reviews. As with standard systematic reviews, there is potential for missed studies, small study bias, missed outcomes, selective reporting bias, and study publication bias, which can influence results and effect the validity of the review [84]. Only 33% of the included systematic reviews deployed a search strategy including one or more technical database (i.e., IEEE or ACM), creating potential for missed studies within the fields of engineering and data science [62,68,69]. To reduce bias between technical vs. medical literature, we included four multi-disciplinary reviewers for the initial selection based on the titles and abstracts and final full-text selection of articles. We also take into consideration that we have a narrow focus within this umbrella review on specific vulnerable groups (older adults, PD, and dementia) and, therefore, may have missed studies that include HAR for younger participants and more general purposes. We further note, however, that by using the narrow scope of older adults, PD, and dementia, we uncover true gaps within this specific and important sector. Lastly, another limitation of this umbrella review was that a meta-analysis was not possible, as the included quantitative literature did not include meta-analyses because measurements were clinically heterogeneous or the authors used inconsistent metrics for analyses.

## 6. Conclusions

HAR has the potential to enrich our knowledge of digital biomarkers, which can help guide clinical decision making for people with PD and dementia. However, for HAR to be sustainable for real-world use in persons with neurodegenerative diseases, such as PD, a foundation of new interdisciplinary collaborations and a culture for technology complemented care is necessary. Researchers should improve the knowledge and understanding of sample size relativity within the field of HAR, meaning that depending on the type of analysis employed, data quantities within smaller sample sizes might be sufficient if there are enough for both the training of models and validation of technologies.

Future perspectives of HAR for monitoring the activities and behavioral symptoms include the application of AI for prediction and observation, strategies for complementing the knowledge and clinical differentiation of health professionals, and the empowerment of research participants, encouraging the ethical inclusion of participants in later stages of disease. Further accountability from the developers of the technologies used for HAR to discourage obsolescence, improve support, and create more research-friendly agreements with universities must be realized for future progress in this field.

## Figures and Tables

**Figure 1 sensors-25-00668-f001:**
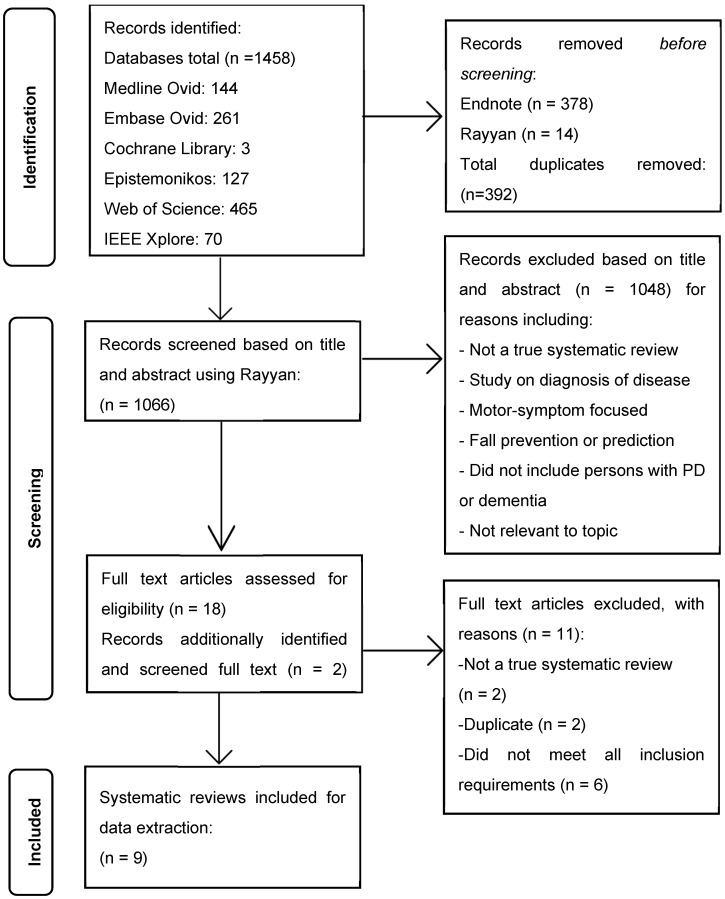
PRISMA flow diagram of study selection process.

**Table 1 sensors-25-00668-t001:** PICO requirements for umbrella review.

Population/Persons	Older Adults with Dementia or PD
**I**ntervention	Human activity and behavior recognition using sensing technology, including wearables and behavioral symptoms and/or functional activities of daily living.
**C**omparison	Current gold standard outcome measures used within the current literature (i.e., Neuropsychiatric Inventory (NPI), Personal Activities of Daily Living (PADL), Polysomnography (PSG), Electrocardiography (EKG), etc.)
**O**utcome	Identification of biomarkers, movement and activity classification models, behavior identification and classification models, measurement methods for activities of daily living, knowledge of basic algorithms and AI used, and information on public datasets specific to human activity recognition (HAR) in older adults with dementia or PD.

**Table 2 sensors-25-00668-t002:** Inclusion and exclusion criteria for umbrella review.

Inclusion Criteria	Exclusion Criteria
Systematic reviews from medical and technical journals	Not a systematic review
Including people with PD or dementia, 65 years or older	People without PD or dementia and younger than 65
Sensing technology used for HAR (including behavioral symptoms): sensors, wearables, radar technology, GPSs, and multimodal sensing systems	Not specific to the management or observation of activities or behaviors, gait specific, motor functions for PD specific, fall specific, apps, diagnosis of disease (early detection), and non-sensor related technology
Literature from the last 5 years (2018–2024)	Published before 2018
English language	Not written in English

## Data Availability

Not applicable.

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
