# Peer review of "Activity and Behavioral Recognition Using Sensing Technology in Persons with Parkinson’s Disease or Dementia: An Umbrella Review of the Literature"

_sensors, 2025, doi:10.3390/s25030668_

Round 1

Reviewer 1 Report

Comments and Suggestions for Authors

The paper presents a Review of papers related to sensing persons with Parkinson's desease or dementia. The paper is well writen and structured but has som flaws related to the references.

A lot of references are located after the full stop of the sentence. For instance, in the first sentence of the Introduction. The reference must be located before the full stop as, in other case, it seems to be part of the next sentence. This is repeated widely across the paper.

In addition, the reference format of MDPI is using square brackets "[ ]" and the paper uses brackets "( )". They must be changed.

The paper also employ a lot of other format references in the form of "Author et al. (year)". This is not the reference format of the journal. Hence, they must be changed. Many times after this reference there is also the numeric reference.

Table 3, as it is very very big, I recommend to repeat the header over every paper analysis to be able to know the meaning of each column.

The content of appendix 5 is located after appendix 6-7. Also, the title of appendix 6-7 is in upper case.

Author Response

Thank you for your helpful comments and suggestions. We appreciate the opportunity to improve the manuscript!

Comment 1: 

A lot of references are located after the full stop of the sentence. For instance, in the first sentence of the Introduction. The reference must be located before the full stop as, in other case, it seems to be part of the next sentence. This is repeated widely across the paper.

In addition, the reference format of MDPI is using square brackets "[ ]" and the paper uses brackets "( )". They must be changed.

Response: Thank you for this helpful correction. I have now reformatted the brackets and corrected the reference placement throughout the manuscript. 

Comment 2: 

The paper also employ a lot of other format references in the form of "Author et al. (year)". This is not the reference format of the journal. Hence, they must be changed. Many times after this reference there is also the numeric reference.

Response 2: Thank you for making me aware of this. We have now revised the sentences including the statement "et al. (year)" to reflect the journals format. 

Comment 3: 

Table 3, as it is very very big, I recommend to repeat the header over every paper analysis to be able to know the meaning of each column.

Response 3: Thank you for your suggestion. In response to yours and the second reviewer's comments, I have attempted to revise Table 3 and have divided the table into three articles per page, over three total pages. The columns and rows have been reversed in an effort for the table to be more easily readable. I will leave the final decision up to the editors as to which format is most appropriate for the journal. We hope this revision will be helpful. 

Comment 4:

The content of appendix 5 is located after appendix 6-7. Also, the title of appendix 6-7 is in upper case.

Response 4: Thank you for noticing this mistake. We have now revised this and it should be correct. 

Reviewer 2 Report

Comments and Suggestions for Authors

 The authors carry out an umbrella review, i.e. a review of reviews, to present the state of the art and the current challenges in methodologies and metrics for use of sensing technologies for the management and monitoring of activities and behavioral symptoms (which is called Human Activity Recognition, HAR) of adults over 65 years of age suffering from Parkinson disease (PD) or dementia.  

It is clear that the population suffering these types of problems is growing as the age of the population increases. It is also true that a lot of effort is being made to use sensors in humans that provide useful information about their health. Therefore, trying to understand whether sensoring is being useful to monitor activities and behavioral symptoms, and how it can be improved, in the benefit of the quality of life of patients, seems to be a meaningful objective.  

In my opinion, the contribution of this work lies in the specificity of the population chosen and the breadth of the aspects considered.  

The research identified nine systematic reviews that fit the chosen inclusion criteria. Several properties and characteristics were extracted from each review: aim and demographics, sensing technologies and observation time, algorithms and artificial intelligence (AI) tools, digital biomarkers, included comparative measures, and results and conclusions.  

The authors devoted a lot of effort to search for relevant reviews of the field and made it in a rigorous manner, so it can be expected that most relevant references have been gathered.  

The main conclusion from the study is that HAR  may provide more knowledge on digital biomarkers which can help in the clinical decision making for patients with PD and dementia. The conclusions are consistent with the results of the study, and they give response to the main questions posed. Moreover, the study concludes that, for this discipline to progress adequately, it is recommended to relativize the importance of the sample size in the studies, as small sample sizes may be sufficient for training models and validate technology, to apply AI for prediction and observation, to develop strategies for complementing knowledge and clinical differentiation of health professionals, to empower participants encouraging ethical inclusion of participants in later stages of disease, to discourage obsolescence and improve support from technological developers, and to foster research friendly agreements with universities.  

I consider that the work has been done correctly, and that it can be useful for the audience of the journal interested in the use of sensing technologies for improving the life of patients suffering from PD or dementia.  

Finally, a couple of minor remarks:  

1) In the caption of Figure 1, it seems that PRISM should be PRISMA.  

2) In the third line of Section 3.2 (line 259 of the manuscript), perhaps the term triaxial sensors could be complemented with that of inertial sensors or IMUs (Inertial Measurement Units).  

3) Table 3 is not particularly easy to read, but probably it is not easy to gather all this amount of information in a more readable way. Perhaps making one table for each review, converting columns to rows, could help.

Author Response

Thank you for your comments and suggestions. I appreciate the opportunity to improve the manuscript. Please see my detailed responses below:

1) In the caption of Figure 1, it seems that PRISM should be PRISMA.

Thank you for this correction. I have now revised the text to correctly reflect the flow chart as "PRISMA". 

2) In the third line of Section 3.2 (line 259 of the manuscript), perhaps the term triaxial sensors could be complemented with that of inertial sensors or IMUs (Inertial Measurement Units).  

Thank you for this helpful suggestion. I have now updated the sentence in line 259-260 to include "inertial sensors". 

3) Table 3 is not particularly easy to read, but probably it is not easy to gather all this amount of information in a more readable way. Perhaps making one table for each review, converting columns to rows, could help.

I really like your suggestion to separate the table into individual articles and/or convert columns to rows. I have included a revised table 3, divided into 3 parts (3 systematic reviews per page - 3 pgs total), and have reversed the columns and rows. I will leave the decision to the editors as to which format is best. I hope this option is helpful and can provide a more easily readable version for the reader.